# Integrating the Idylla™ System Alongside a Real-Time Polymerase Chain Reaction and Next-Generation Sequencing for Investigating Gene Fusions in Pleural Effusions from Non-Small-Cell Lung Cancer Patients: A Pilot Study

**DOI:** 10.3390/ijms25147594

**Published:** 2024-07-11

**Authors:** Stefania Scarpino, Alvaro Leone, Dino Galafate, Francesco Pepe, Umberto Malapelle, Sandra Villani, Enrico Giarnieri, Giulio Maurizi, Claudia De Vitis, Rita Mancini, Massimiliano Mancini, Arianna Di Napoli, Andrea Vecchione, Emanuela Pilozzi

**Affiliations:** 1Department of Clinical and Molecular Medicine, Morphologic and Molecular Pathology Unit, St. Andrea University Hospital, Sapienza University of Rome, 00189 Rome, Italy; dino.galafate@uniroma1.it (D.G.); svillani@ospedalesantandrea.it (S.V.); enrico.giarnieri@uniroma1.it (E.G.); claudia.devitis@uniroma1.it (C.D.V.); rita.mancini@uniroma1.it (R.M.); mamancini@ospedalesantandrea.it (M.M.); arianna.dinapoli@uniroma1.it (A.D.N.); andrea.vecchione@uniroma1.it (A.V.); emanuela.pilozzi@uniroma1.it (E.P.); 2Anatomic Pathology Unit, San Camillo-Forlanini Hospitals, 00152 Rome, Italy; aleone@scamilloforlanini.rm.it; 3Department of Public Health, University of Naples Federico II, 80138 Naples, Italy; francesco.pepe4@unina.it (F.P.); umbertomalapelle@gmail.com (U.M.); 4Department of Thoracic Surgery, Sant’ Andrea Hospital, Sapienza University of Rome, 00189 Rome, Italy; giulio.maurizi@uniroma1.it

**Keywords:** malignant pleural effusion, NSCLC, Idylla™ system, gene fusion, cfRNA

## Abstract

Malignant pleural effusion (MPE) from patients with advanced non-small-cell lung cancer (NSCLC) has been proven valuable for molecular analysis; however, simultaneous detection of driver fusions in MPE is still challenging. In this study, we investigated the Idylla™ GeneFusion Panel, a stand-alone test in tissue samples, in the evaluation of *ALK*, *ROS1*, *RET* and *MET* ex14 skipping mutations in MPE and compared its performance with routine reference methods (Real-time-based and Next-generation Sequencing—NGS). The inclusion criteria for sample selection were as follows: advanced NSCLC harboring *ALK*, *ROS1*, *RET* fusions or *MET* exon-skipping alterations and the availability of MPE collected at diagnosis or disease progression. Molecular alterations have been investigated on tissue by fluorescence in situ hybridization (FISH) or Real-time PCR or NGS. For molecular profiling with the Idylla™ GeneFusion, 200 µL of MPE supernatants combined with 50 µL of RNA Later solution were loaded into the Idylla™ cartridge without cfRNA extraction. The Idylla™ GeneFusion Assay performed on MPEs was able to confirm molecular profile, previously diagnosed with conventional methods, in all cases. Our data confirm that MPE are suitable material for investigating fusion alterations. The Idylla™ GeneFusion, although indicated for investigation of tissue samples, offers the possibility of performing a molecular characterization of supernatants without undertaking the entire cfRNA extraction procedure providing a rapid and reliable strategy for the detection of actionable genetic alterations.

## 1. Introduction

Precision medicine has radically modified the clinical paradigm for lung cancer patients [1,2]. In particular, non-small-cell lung cancer patients (NSCLC) with adenocarcinoma histological type (ADC) dramatically respond to target drugs, improving clinical outcomes when tumor cells harbor hotspot druggable molecular alterations [3,4,5]. As a consequence, an impressive number of predictive biomarkers need to be tested to stratify NSCLC patients clinically [6] in the clinical administration of NSCLC patients; among them, it is mandatory to test a heterogeneous landscape of molecular alterations, including single nucleotide variations (SNVs), small insertions/deletions (indels) and aberrant fusion transcripts. Noteworthy diagnostic specimens from advanced NSCLC patients are often affected by inadequate material for morphological evaluation and molecular approaches [7,8,9,10]. In this scenario, a liquid biopsy that covers all the minimally invasive diagnostic procedures for recovering nucleic acids from biological fluids (peripheral blood, pleural effusion, sputum, urine) emerged as an integrative tool for the molecular profiling of predictive biomarkers [10]. In particular, malignant pleural effusion (MPE) occurs in a not negligible percentage of novel diagnosed lung cancer patients (ranging from 15.0 to 50.0%) [11]. MPE is routinely considered a gold standard diagnostic approach for pleural invasion [11,12]. Moreover, previous studies demonstrated the abundance of circulating tumor nucleic acids (ctDNA) in MPE samples detectable by molecular techniques [13,14]. Conversely, circulating tumor RNA (cfRNA)-based testing strategies have been proven to be more difficult for the high rate of degradation of RNA and require optimized workflow in sample collection, nucleic acids management and testing strategies to evaluate clinically relevant aberrant transcripts in MPE specimens [15]. The Idylla^TM^ GeneFusion Panel, a rapid, efficiently managing and fully automatized RT-PCR approach, was set up to detect aberrant fusions (*ALK*, *ROS*, *MET* and *RET*) starting from tumor tissue specimens [16,17,18].

Here, we investigated the technical feasibility of the Idylla^TM^ GeneFusion Panel (Biocartis NV, Mechelen, Belgium) to detect aberrant fusion transcripts in MPE supernatants from NSCLC patients.

## 2. Results

### 2.1. MPE Molecular Profiling with Standard Methods

A median cfRNA of 34 ng/µL (ranging from 6.5 to 99.0 ng/µL) was obtained from MPE (Table 1). Overall, cfRNAs MPE samples were successfully analyzed by RT-PCR or NGS in all samples (9/9, 100%). The median percentage of tumor cells in MPE smears was 52.0% (ranging from 5.0 to 95.0%) (Table 1).

### 2.2. Results of the Calibration Curve

A dilution study on MPE sample #9 that harbors RET fusion was conducted to establish the lowest cfRNA concentration and the minimum cell content for specific fusion detection by the Idylla™ System. As shown in Table 2, Idylla™ was able to identify the specific fusion transcript starting from cfRNA 18 total nanograms and until 2.0% of neoplastic cell content.

### 2.3. Performance of the Idylla Gene Fusion Assay on MPE Supernatants and Cytological Smears

In 9 out 9 cases in which MPE was directly analyzed by the Idylla™ GeneFusion Panel (Biocartis NV, Mechelen, Belgium), the molecular profile was confirmed, previously diagnosed on histological tissue and cfRNA extracted from MPE. Notably, Idylla™ identified fusions or imbalances (6/9 and 3/9, respectively) in MPE supernatant samples (Table 1).

In 5 out of 6 (83%) cytological smears directly scraped into the Idylla™ cartridge, the Idylla™ (Biocartis NV, Mechelen, Belgium) platform identified fusions (Table 1).

## 3. Discussion

In the era of personalized medicine, molecular profiling has been proven mandatory for the clinical administration of NSCLC patients. MPE samples are considered the most suitable clinical tool for diagnosing and staging advanced lung cancer patients [12]. If technical approaches easily support cfDNA molecular analysis from biological fluids, including MPE, low stability and high fragmentation rate severely impact cfRNA implementation in clinical practice [17]. In this scenario, optimized technical approaches may improve the molecular profiling of cfRNA samples for NSCLC patients. To our knowledge, this is the first study that investigates the possibility of identifying gene fusion in MPE of NSCLC patients using the Idylla™ system. Idylla™ performed on unextracted supernatant from MPE demonstrating a molecular profile overlap (9 out 9 cases) with reference methods (RT-PCR and NGS).

MPE has already been proven to be a valuable source of circulating tumor-derived nucleic acids detectable with routine molecular techniques such as Real-time PCR and NGS; however, our data demonstrated that MPE can be used directly overcoming technical procedures of nucleic acid extraction. This procedure reduces turnaround time to <2 h.

The inadequacy of neoplastic material to perform molecular analysis of predictive biomarkers is one of the most critical issues in molecular profiling of solid tumor patients [18]. A challenging point regards the identification of minimum cfRNA starting input to detect aberrant transcripts. One notable aspect of this study is that the successful analysis of MPE specimens was not affected by neoplastic cell percentage (range 7.0–88.0%).

The longitudinal series of diluted samples from patient #9 provided valuable insights into the analytical sensitivity of the Idylla™ system, showing valuable technical sensitivity for detecting target molecular alterations starting from 2% of neoplastic cell percentage and with a minimum RNA quantity of 18 ng. Our data are concordant with the study of Buglioni et al., which found 20 ng of total RNA to be the lowest threshold for the Idylla™ GeneFusion Assay [17]. These findings could optimize sample processing protocols to maximize the detection rate of gene fusions in MPE samples with varying levels of tumor cellularity.

MPE is a precious source of neoplastic cells. They are obtained with centrifugation and used to prepare smears for cytological diagnosis and/or cell block formalin-fixed paraffin embedding. In clinical practice, molecular testing based on pleural effusion cytology is highly recommended by clinical guidelines [16]. In this study, cells scraped from already stained smears of 6 patients were analyzed with Idylla™, and the specific molecular alteration was identified in all cases but 1 sample (17%) with a low, although not the lowest, tumor cells percentage (7%). Since the smear was rich in inflammatory cells, we can suppose that it could have affected the results. The small number of samples investigated, however, does not allow us to draw conclusions. Our observation strengthens the data reported by Buglioni et al. that indicated a limit of 5% neoplastic cellularity to obtain a valid result on smears with the Idylla™ GeneFusion Assay [17].

Some limitations in this study could be the object of further investigations. Since it was a retrospective collected cohort, it was not possible to evaluate if delayed MPE processing could affect the results. MPEs were collected from routine practice and the exact lap time from thoracocentesis to analysis was not recorded. Secondly, since our laboratory is a referral center for molecular evaluation of biofluids (plasma, pericardial and pleural effusions), treatment data were unavailable for all patients. Lastly, although our results have proven high concordance with the reference method applied in molecular analysis of MPEs as RT-PCR and NGS, a larger number of samples collected prospectively could strengthen the findings.

Overall, this is the first study that demonstrates the possibility of conducting a molecular characterization of supernatants without having to undertake the entire cfRNA extraction procedure. This can undoubtedly represent a time advantage in providing a rapid and effective response to the choice of therapy for NSCLC patients. Integrating Idylla™ alongside established molecular techniques in MPE presents a promising tool for enhancing the molecular profiling of non-small-cell lung cancer (NSCLC) patients, offering rapid and reliable detection of actionable genetic alterations to guide personalized treatment decisions for NSCLC patients.

## 4. Materials and Methods

### 4.1. Patients

A series of nine advanced ADC patients with MPE, enrolled by St. Andrea University Hospital between April 2022 and December 2023, were selected. In 8 patients, the diagnostic sample was represented by biopsy (bronchus or pleura) and in 1 patient by surgical resection of the right superior lobus (pathological stage IIIB at diagnosis). Patients were previously tested on tissue specimens for RNA-based molecular alterations (*ALK*, *ROS1*, *RET* fusions and *MET* ex14 skipping alterations). *ALK*, *ROS1* and *RET* aberrant rearrangements and *MET* ex14 skipping molecular alteration were detected in 5 out of 9 (55.6%), 2 out of 9 (22.2%), 1 out of 9 (11.1%) and 1 out 9 (11.1%), respectively. Reference methods for detecting gene fusions in tissue samples included immunohistochemistry (IHC), fluorescence in situ hybridization (FISH), reverse transcription–polymerase chain reaction (RT-PCR) and Next-generation Sequencing (NGS). Immunostaining was performed using an automated immunostainer platform (BOND III Leica Biosystems, Seoul, Republic of Korea) with the following antibodies: ALK clone D5F3 and ROS1 clone D4D6, as recommended by manufacturer protocol. FISH for both ALK (Kreatec dual-color break apart) and ROS1 (Kreatec dual-color break apart) was performed using an automated immunostainer platform (BOND III Leica Biosystems). *ALK*, *ROS*, *RET* gene fusions and *MET* ex14 skipping were investigated by RT-PCR (LUNG-RT48, ENTROGEN, Woodland Hills, CA, USA) or NGS (Oncomine Focus Assay, Thermo Fisher, Waltham, MA, USA) as recommended in the manufacturer’s instructions.

### 4.2. Standard Technical Management of MPE Samples

An amount of 2 out of 9 (23%) MPEs were collected at diagnosis and 7 out of 9 (77%) at recurrence. According to the standardized workflow, MPE samples were centrifuged at 1500 rpm for 10 min to separate the supernatant from the cell pellet. The cell pellet was smeared and stained with Papanicolau to evaluate the neoplastic cell percentage. A total of 4 mL of supernatant was dedicated to cfRNA purification, adopting the Promega Maxwell^®^ (Madison, WI, USA) automatic system (AS1840 RSC ccfDNA LV and AS1680 ccfRNA plasma kit) following manufacturer procedures. cfRNA fragmentation and concentration were evaluated by Real-time PCR (RNA quantification strips Myriapod^®^ NGS Cancer panel RNA AMP LAB, Diatech Pharmacogenetics srl Jesi (AN)–Italy). cfRNAs were tested for molecular alterations by RT-PCR (LUNG-RT48, ENTROGEN) or NGS (Oncomine Lung Cell-Free Total Nucleic Acid Research Assay or Oncomine Focus Assay, Thermo Fisher).

### 4.3. MPEs Investigation by Idylla™ Platform Cartridge Genefusion Panel

In order to investigate the potential application of the Idylla^TM^ Platform for molecular profiling of MPEs 150–200 µL supernatants of MPE, combined with 50 µL of RNA Later solution, were loaded into Idylla™ cartridge without cfRNA extraction. In parallel, cells from cytological smears were directly scraped into an Idylla™ cartridge to detect molecular alterations (Figure 1).

### 4.4. Calibration and Standardization Curves

To determine the minimum MPE cfRNA amount required to successfully evaluate molecular alteration, serial dilution of an exemplificative case (sample #9) harboring the RET fusion was set up. Of note is that serial dilution points were obtained from total pleural effusion (not centrifuged) containing the tumor cells. Briefly, the supernatant was serially diluted (A–H). An amount of 200 µL of each dilution point was loaded into the Idylla^TM^ cartridge. The cfRNA amount in each dilution (from 250.0 ng to 8.0 ng) was quantified with a NanoDrop™ Spectrophotometer (Thermo Scientific™) following standardized manufacturer procedures. The same approach was adopted to count tumor cells across serial dilution points by using a Burker chamber. Briefly, 10 µL of each dilution was combined with 10 μL of Trypan blue (diluted 1:2 in PBS) and transferred in the Bürker chamber. Cells were counted in at least three fields and averaged, and the number of cells was obtained following this algorithm: N° cells × mL = (N° cells/N° quadrants) × 200 × 1000.

The Horizon *ALK-RET-ROS1* Fusion FFPE RNA Reference Standard (HD784) was used to create a standard curve to test the system’s sensitivity. The Horizon standard curve showed that the Idylla system could identify fusions by loading at least 15 ng of RNA (Appendix A).

## Figures and Tables

**Figure 1 ijms-25-07594-f001:**
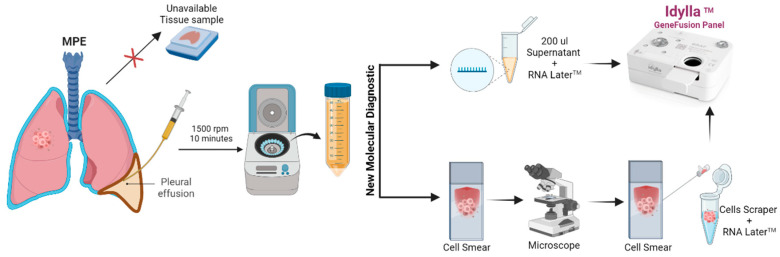
Potential algorithm that proposes the integration of the Idylla^TM^ Gene Fusion Panel cartridge in the evaluation of molecular alterations in pleural effusions of patients with NSCLC. (Permission was granted for Biorender the image.)

**Table 1 ijms-25-07594-t001:** Patient characteristics and molecular results obtained with Idylla™ GeneFusion and reference method in the assessment of *ALK*, *ROS1*, *RET* translocations and *MET* ex14 skipping.

Case n.	Sex/Age	Histological Type	Tissue Alteration RTPCR/NGS	MPE %Tumor Cells *	MPE/cfRNA Conc. (ng/µL)	MPE/cfRNA RTPCR°/NGS§	MPE/Idylla	Idylla Cytological Smear **
1	M/54	ADK	*ALK+*	7.0%	8	*ALK+°§*	*ALK+@*	*ALK−*
2	M/85	ADK	*RET+*	95.0%	99	*RET+°§*	*RET+*	*RET+*
3	M/63	ADK	*ALK+*	5.0%	38	*ALK+°§*	*ALK+*	*ALK+*
4	M/64	ADK	*ALK+*	60.0%	10.4	*ALK+°§*	*ALK+@*	*ALK+*
5	F/64	ADK	*ALK+*	60.0%	17	*ALK+°§*	*ALK+@*	*ALK+*
6	M/64	ADK	*ALK+*	50.0%	89	*ALK+°§*	*ALK+*	*ALK+*
7	F/70	ADK	*ROS1+*	50.0%	11.2	*ROS1+°§*	*ROS1+*	*N.A.*
8	M/65	ADK	*Met skip+*	*N.A.*	*N.A.*	*N.A.*	*Met skip+*	*N.A.*
9	F/76	ADK	*RET+*	88.0%	6.5	*RET+°§*	*RET+*	*N.A.*

Abbreviations: M: male; F: female; ADK: adenocarcinoma; +: positive; −: negative; N.A.: not applicable; * % was evaluated in cytological smear sample obtained from MPE; ** cells from MPE/smears were scraped and loaded on the Idylla cartridge. @ presence of a 5′-3′ expression imbalance of the gene. § NGS, ° RTPCR

**Table 2 ijms-25-07594-t002:** Idylla™ GeneFusion performance in a MPE dilution experiment.

MPE Case n.9	* Dil.A	Dil.B	Dil.C	Dil.D	Dil.E	Dil.F	Dil.G	Dil.H
** cfRNA ng/200 µL	250 ng	175 ng	139.5 ng	105.3 ng	65.7 ng	30.2 ng	18 ng	8 ng
Cells/mL	39 × 10^4^	33 × 10^4^	17 × 10^4^	10 × 10^4^	5 × 10^4^	3 × 10^4^	1 × 10^4^	ND
% Cells	88%	74%	38%	22%	11%	6%	2%	ND
Cq *RET* ***	*RET+* 26.3	*RET+* 26.5	*RET*+ 31.1	*RET*+ 31.5	*RET*+ 32.1	*RET*+ 33.5	*RET*+ 35.4	*RET*− ND

* Serial dilution (1:1) from the starting sample (Dil A). The starting sample corresponds to 200 microliters of MPE supernatant. ** cfRNA amount is contained at each point of the dilution. *** Quantification cycle (Cq) at which the Idylla™ GeneFusion cartridge detected the *RET* signal. Abbreviations: ND—not detected.

## Data Availability

All data generated or analyzed during this study are included in this published article (and its Appendix A).

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
