# Peer review of "Integrating the Idylla™ System Alongside a Real-Time Polymerase Chain Reaction and Next-Generation Sequencing for Investigating Gene Fusions in Pleural Effusions from Non-Small-Cell Lung Cancer Patients: A Pilot Study"

_ijms, 2024, doi:10.3390/ijms25147594_

Round 1

Reviewer 1 Report

Comments and Suggestions for Authors

The manuscript by Stefania Scarpino et al, titled Integrating IdyllaTM System alongside RT-PCR and NGS for investigating gene fusions in pleural effusions from NSCLC patients: a pilot study. This study provides valuable insights into investigating the use of Idylla™ GeneFusion Panel for detecting specific genetic mutations in malignant pleural effusion (MPE) samples from patients with advanced non-small cell lung cancer (NSCLC). I recommend that the manuscript in its current form is not ready to be published,

Point-by-point comments to improve:

1. While the abstract mentions using 200μl supernatants combined with 50μl of RNA Later solution, it lacks detailed methodological explanations, such as the criteria for selecting the patients and specific conditions under which the assays were performed.

2. There are minor grammatical errors and awkward phrasing that could be improved for better readability. For example, "molecular profile, previously diagnosed on histological tissue and on cfRNA extracted from MPE with conventional methods, by analyzing unextracted cfRNAs MPE in all cases" is convoluted.

3.  The study involved only 9 patients, which is a very small sample size. This limits the generalizability and statistical power of the findings.

4.   The introduction lacks a clear, logical flow. It jumps between topics, such as the importance of molecular profiling, the role of NGS and RT-PCR, and the challenges with diagnostic specimens, without smooth transitions.

5. There are instances of redundant information and convoluted sentences that can be simplified for better readability.

6. While mentioning the Idylla™ platform and its capabilities, the text lacks a concise explanation of why this study is necessary and what gap it aims to fill.

7.  The results section is somewhat disorganized and lacks a logical flow. It jumps between different findings without clear transitions.

8.  There is some redundant information, such as repeatedly stating the confirmation of molecular profiles by the Idylla™ platform.

9.  The section mentions a failed molecular analysis but does not provide enough detail on the circumstances or potential reasons for this failure.

8.  Some points are repeated, which detracts from the overall clarity and conciseness.

9.  There is insufficient detail on the one sample where the molecular alteration was not identified. This could be explored further to understand the limitations of the Idylla™ platform.

10.  The use of technical jargon without adequate explanation may make the discussion less accessible to a broader audience.

11. The discussion occasionally lacks smooth transitions between points, making it harder to follow the logical flow of the argument.

Comments on the Quality of English Language

There are minor grammatical errors and awkward phrasing that could be improved for better readability.

Author Response

Reviewer 1

Point-by-point comments to improve:

  1. While the abstract mentions using 200μl supernatants combined with 50μl of RNA Later solution, it lacks detailed methodological explanations, such as the criteria for selecting the patients and specific conditions under which the assays were performed.

Reply: We added in the abstract more detailed methodological explanations

  1. There are minor grammatical errors and awkward phrasing that could be improved for better readability. For example, "molecular profile, previously diagnosed on histological tissue and on cfRNA extracted from MPE with conventional methods, by analyzing unextracted cfRNAs MPE in all cases" is convoluted.

Reply: We improved the quality of the test simplifying convoluted sentences.

  1. The study involved only 9 patients, which is a very small sample size. This limits the generalizability and statistical power of the findings.

Reply: We are aware that our cohort is small however we would like to underlined that it is of great value as fusions in NSCLC are rare events and for the purpose of this study it was mandatory to have tissue biopsy for molecular profiling with standard methods (FISH, Real Time PCR/ NGS) and malignant pleural effusion to be tested with the same methods plus Idylla.

  1. The introduction lacks a clear, logical flow. It jumps between topics, such as the importance of molecular profiling, the role of NGS and RT-PCR, and the challenges with diagnostic specimens, without smooth transitions.

Reply: We modified the text and, hopefully, made it smoothly.

  1. There are instances of redundant information and convoluted sentences that can be simplified for better readability.

Reply: We modified the text and, hopefully, made it smoothly.

  1. While mentioning the Idylla™ platform and its capabilities, the text lacks a concise explanation of why this study is necessary and what gap it aims to fill.

Reply: In the discussion section we stated the advantage of using Idylla for fusion identification without cfRNA extraction

  1. The results section is somewhat disorganized and lacks a logical flow. It jumps between different findings without clear transitions.

Reply: We modified the sequence of the results paragraph in order to make it more linear and smooth.

  1. There is some redundant information, such as repeatedly stating the confirmation of molecular profiles by the Idylla™ platform.

Reply: We simplified the text in the discussion section.

  1. The section mentions a failed molecular analysis but does not provide enough detail on the circumstances or potential reasons for this failure.

Reply: In this work we document that the Idylla test identifies fusions in all analyzed MPE samples (9/9). In one of the 6 smears analyzed the expected fusion was not identified. It was a case with low neoplastic cellularity and a rich lymphocyte component. We have inserted a comment in the discussion.

  1. Some points are repeated, which detracts from the overall clarity and conciseness.

      Reply: We have simplified and made the text more concise

  1. There is insufficient detail on the one sample where the molecular alteration was not identified. This could be explored further to understand the limitations of the Idylla™ platform.

Reply: See point 9

  1. The use of technical jargon without adequate explanation may make the discussion less accessible to a broader audience.

Reply: We simplified the discussion to make it more accessible

  1. The discussion occasionally lacks smooth transitions between points, making it harder to follow the logical flow of the argument.

Reply: See point 12

Reviewer 2 Report

Comments and Suggestions for Authors

Dear Editor and Authors,

I would like to express my gratitude for the opportunity to review the manuscript titled "Integrating IdyllaTM System alongside RT-PCR and NGS for Investigating Gene Fusions in Pleural Effusions from NSCLC Patients: A Pilot Study."

This is an innovative study with significant clinical relevance. I have only a few minor suggestions to improve the manuscript. While the topic is highly relevant, the patient demographics (E.g.: age) and disease stage/location information are missing and should be expanded upon in the methods section.

The discussion can benefit from a more explicit analysis of the variability in tumor cell percentages and more detailed comparisons with other diagnostic platforms.

The limitations section should address the low number of patients, which may limit the generalizability of the findings and the external validity.

Figure 1 would fit better in the methods.

Some typographical mistakes can be found. For example, in Line 77: “26,27.”

Author Response

Reviewer 2

Point-by-point comments to improve:

  1. This is an innovative study with significant clinical relevance. I have only a few minor suggestions to improve the manuscript. While the topic is highly relevant, the patient demographics (E.g.: age) and disease stage/location information are missing and should be expanded upon in the methods section.

Reply: In the second column of  table 1, we indicated patient age. In the method section we added information on bioptic samples and stage.

2. The discussion can benefit from a more explicit analysis of the variability in tumor cell percentages and more detailed comparisons with other diagnostic platforms.

Reply: Comparative studies between Idylla and standard analysis methods for the identification of fusions in NSCLC have already been published. The aim of our work was to evaluate the possibility of using Idylla gene fusion, a tissue-based test, for evaluation of fusions also in supernatant MPEs

3. The limitations section should address the low number of patients, which may limit the generalizability of the findings and the external validity.

Reply: We added a comment on the small size of our cohort in the limitations section

4. Figure 1 would fit better in the methods.

Reply: We moved Figure 1 in the method section.

5. Some typographical mistakes can be found. For example, in Line 77: “26,27.”

Reply: We corrected typing mistakes.

Round 2

Reviewer 1 Report

Comments and Suggestions for Authors

Thank you for addressing the issues I raised.

The manuscript can be published.